# The Dynamics of (Not) Unfollowing Misinformation Spreaders

## ABSTRACT

Many studies explore how people "come into" misinformation exposure. But much less is known about how people "come out of" misinformation exposure. *Do people organically sever ties to misinformation spreaders? And what predicts doing so?* Over six months, we tracked the frequency and predictors of ~1M followers unfollowing ~5K health misinformation spreaders on Twitter. We found that misinformation ties are persistent. Monthly unfollowing rates are just 0.52%. Users are also 31% more likely to unfollow *non*-misinformation spreaders than they are to unfollow misinformation spreaders. Although generally infrequent, the factors most associated with unfollowing misinformation spreaders are (1) redundancy and (2) ideology. First, users initially following many spreaders, or who follow spreaders that tweet often, are most likely to unfollow later. Second, liberals are more likely to unfollow than conservatives. Overall, we observe strong persistence of misinformation ties. The fact that users rarely unfollow misinformation spreaders suggests a need for external nudges and the importance of preventing exposure from arising in the first place.

## CCS CONCEPTS

• **Human-centered computing** → **Social networks**; **Social media**; **Computer supported cooperative work**; **Social network analysis**.

## KEYWORDS

misinformation, social media, unfollowing, responsible web, twitter, social networks

**ACM Reference Format:**
Anonymous Author(s). 2018. The Dynamics of (Not) Unfollowing Misinformation Spreaders. In *Proceedings of Make sure to enter the correct conference title from your rights confirmation emai (Conference acronym 'XX)*. ACM, New York, NY, USA, 11 pages. https://doi.org/XXXXXXX.XXXXXXX

## 1 INTRODUCTION

Misinformation exposure affects key decisions like compliance with health regulations [11] and vaccine uptake intention [28]. Due to its importance, many studies explore how users become exposed to misinformation. But much less is known about when users choose to *unfollow* misinformation sources. Even some of the most *fundamental* empirical questions around misinformation unfollowing are still largely unanswered. For example, is unfollowing misinformation sources generally common or rare? And is misinformation

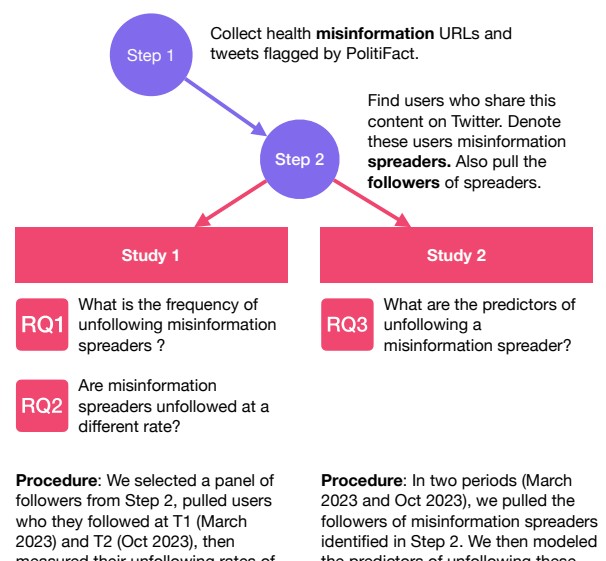

**Figure 1: Graphical summary of studies.**

exposure *self-reinforcing* or *self-correcting*? That is: Is high initial exposure predictive of *higher* or *lower* unfollowing rates?

Little is known about misinformation unfollowing—but understanding its frequency and predictors benefits researchers and platforms interested in a more responsible Web. First, regarding frequency: If unfollowing is rare, then this suggests content moderation and interventions to stop connections from forming in the first place are crucial for stopping exposure. But if unfollowing is common, then perhaps users organically reduce their misinformation exposure—reducing the content moderation burden. If unfollowing is common, this also complicates the interpretation of studies describing a 'snapshot' of misinformation exposure at a moment in time. Second, understanding the predictors of unfollowing misinformation spreaders can help both researchers and platforms design interventions to *further* increase unfollowing.

Here we provide the first large-scale account of misinformation unfollowing. We modeled the frequency and predictors of Twitter users (~1M followers, ~5K spreaders, ~3M edges) unfollowing health misinformation spreaders. We identified misinformation spreaders as Twitter users who shared content that PolitFact flagged as health misinformation. The study period ran from March 2023 to October 2023. We focus on health misinformation for two reasons. First, claims related to health are often more straightforwardly falsifiable than claims in other domains. Second, the cost of false health misinformation can be high.

Across two studies, we answer three research questions related to the frequency and predictors of unfollowing misinformation spreaders. See Figure 1 for a graphical overview.

**RQ1 (Study 1):** How often are misinformation spreaders unfollowed? *We track how often spreaders are unfollowed from the first time point to the second time point.*

**RQ2 (Study 1):** Are misinformation spreaders unfollowed at a different rate than non-misinformation spreaders? *We compared how often a subset of followers unfollows spreaders vs non-spreaders.*

**RQ3 (Study 2):** What predicts unfollowing a misinformation spreader? *We model unfollowing as a function of initial exposure, ideology, edge characteristics, and platform activity.*

## 2 RELATED WORK

Here we review related work on (1) factors perpetuating misinformation exposure and (2) predictors of unfollowing on social media.

### 2.1 Misinformation Exposure

Many studies explore how misinformation exposure arises in the first place. There is no uniform pathway. Factors perpetuating misinformation exposure can be grouped into two buckets: *individual* (e.g., ideology, cognitive reflection, attention) and *environmental* (e.g., algorithms, defaults).

*2.1.1 Individual Factors.* Misinformation exposure is driven by individual-level factors, such as selective exposure [8] cognitive reflection, [25], and inattention to accuracy [23].

*Selective Exposure.* People consume (mis)information consistent with their ideology. In a large study based on browsing data, conservatives consumed more untrustworthy news content than liberals overall [8] in the 2016 election. But for both Trump and Clinton supporters, users were more likely to visit untrustworthy websites consistent with their political ideology—with an especially large effect for Trump supporters [8]. Another large-scale study also supports selective exposure: people are much less likely to click on cross-ideology links in newsfeeds [1]. Moreover, selective exposure appears to be a stronger driver of misinformation consumption for users with extreme political ideologies. In 2016, consumption of fake news with respect to ideology was 'v-shaped': both extreme liberals and extreme conservatives consumed larger amounts of fake news [19]. Although fake news exposure decreased during the 2020 election, it was still the case that conservatives—and particularly, extreme conservatives—consumed more fake news than liberals [19]. Robertson et al. compared the amount of unreliable and partisan news users were exposed to in search results versus the amount of unreliable and partisan news they consumed. Consumption of the latter was significantly higher than the former. This also suggests users are *seeking* unreliable news (over and above what is being shown to them by platforms). Overall, there is large-scale evidence that selective exposure drives misinformation consumption.

*Cognitive Reflection.* Cognitive reflection is implicated in misinformation exposure and belief. The Cognitive Reflection Test (CRT) measures a person's tendency towards analytical thinking [7]. People who are less likely to engage in such thinking are more likely to consume and believe misinformation. In lab experiments, low-CRT participants are more likely to believe conspiracy theories and fake news [24, 25]. CRT correlates with on-platform behavior [20]: Twitter users lower in CRT are more likely to share low-quality news, and there are differences in Twitter accounts that low vs. high CRT users follow.

*Inattention.* Users may *share* misinformation because they are not paying attention to the accuracy of what they share [23, 26]. This line of research is principally concerned with *sharing* of misinformation but also has implications for *exposure* to misinformation. Users likely encounter more misinformation in their feeds because other users share content inattentively.

*2.1.2 Environmental Factors.* Misinformation exposure also may arise more incidentally through environmental factors. Here, we refer to both environments built for users (e.g., social media algorithms) and environments users build for themselves (e.g., defaults).

*Algorithms.* There is mixed evidence regarding the role of recommendation algorithms in promoting misinformation. Several auditing studies found consuming misinformation content leads to more misinformation content being recommended by algorithms [10, 21, 30]. These studies suggest recommendation algorithms may amplify *already-existing* misinformation exposure. Yet, the largest field experiment to date on this topic suggests algorithms *do not* promote misinformation on Facebook and Instagram [9]. In a different domain (extremist content on Youtube), Chen et al. also questions the significance of algorithmic effects and points to demand effects as a more important mechanism.

*Defaults.* Misinformation exposure can arise through the defaults that users set. Users curate a set of sources they regularly consume, what Kim calls a 'media repertoire'. This repertoire can serve as a default filter for news. For example, Flaxman et al. showed that most online news consumption was driven by users visiting their homepage. That is, the 'defaults' that users created (i.e., setting a homepage) strongly influenced the content that users saw. Analogously, the content from the people that a user chooses to follow online can be thought of as the 'default' content that the user sees. Combined with homophily, this can create ideologically segregated filter bubbles or echo chambers [3, 5, 29]. This is a different dynamic than selective exposure, where one is seeking out the information itself. Of course, the initial choosing of defaults is an *individual-level* decision. But this decision then creates a content *environment* that may expose one to misinformation.

### 2.2 Unfollowing on Social Media

Several variables have emerged as key predictors of unfollowing. A reciprocal tie between Twitter users A and B is associated with significantly lower odds of A unfollowing B [15, 16, 32]. Reciprocity may be a cause or effect of tie strength. Kwak et al. argues that reciprocal relationships on social media cause emotional closeness between two users since they see each other's posts. Alternatively, reciprocity may signal two users are already friends or acquaintances in an offline context [18]. Additionally, *redundancy*—either similar content to what the user follows or burst-tweeting—is a predictor of unfollowing [15, 17]. More relevant, Kaiser et al. found participants reported higher theoretical intentions to unfollow or

**Table 1: How samples map to studies and research questions.**

| Sample Name | Description | Used in Study | Answers Research Question(s) |
|---|---|---|---|
| Initial Sample | Initial pull of spreaders and followers | - | - |
| Modeling Sample | Subset of the Initial Sample used for modeling predicting unfollowing | Study 2 | RQ3 |
| Rate Sample | Subset of the Modeling Sample used for rates of unfollowing | Study 1 | RQ1, RQ2 |

block an imagined cross-party friend for posting misinformation, and Yoo et al. found liberals were more likely to report unfriending.

## 3 SPECIFIC HYPOTHESES

Considering unfollowing of misinformation spreaders in particular, we are specifically interested in two key predictors: T1 exposure (the number of spreaders a follower follows at time point 1) and partisan ideology. These variables represent network and individual-level variables, respectively.

### 3.1 Initial (T1) Exposure Effect on Unfollowing

It is not clear, based on the existing literature, if T1 exposure (the number of misinformation spreaders, hereafter 'spreaders', followed at the first time point) should be positively or negatively related to unfollowing. And yet, this empirical relationship is important since it can speak to the extent to which misinformation exposure is self-correcting or self-reinforcing:

- **Reversion Hypothesis (H1)**: Higher exposure at T1 is associated with *higher* unfollowing at T2—meaning misinformation is partially *self-correcting*.
- **Inertia Hypothesis (H2)**: Higher exposure at T1 is associated with *lower* unfollowing at T2—meaning misinformation is partially *self-reinforcing*.

The logic for the *reversion* hypothesis **(H1)** is that high T1 exposure would signal high redundancy. And if misinformation ties are redundant, the probability of each being unfollowed should increase. Additionally, if the follower came to be exposed to misinformation through some incidental mechanism and not an intentional one, then this makes regression to the mean a likely prediction. Both mechanisms would suggest *increased* unfollowing in T2 if T1 exposure is high.

The logic for the *inertia* hypothesis **(H2)** is that high misinformation exposure at T1 may be (1) a *consequence* of selective exposure due to extreme ideology [8] or (2) a *cause* of believing in the misinformation (since exposure[1] to misinformation increases its perceived accuracy [22]). Both mechanisms would suggest *decreased* unfollowing in T2 if T1 exposure is high.

### 3.2 Partisan Ideology Effect on Unfollowing

We hypothesized that ideology (left/right, moderate/extreme) would affect if a user unfollowed a misinformation spreader. Some evidence suggests liberals have a higher unfollowing/unfriending rate [33] so we hypothesized that **(H3)** liberals may be more likely to unfollow here (though political unfollowing is a different phenomenon). We also hypothesized that **(H4)** politically extreme users

would be less likely to unfollow since ideological extremity is correlated with misinformation exposure [19]. But in 2020, *extreme liberals* decreased consumption of fake news [19]. And even in 2016, the top decile of conservatives had larger misinformation consumption than the top decile of liberals [8]. Consequently, we hypothesized that there would be a negative interaction effect **(H5)** between conservatism and ideological strength; an equivalent increase in ideological strength would reduce the probability of unfollowing more for conservatives than it would for liberals. We were also interested in **(Q)** a possible interaction between T1 exposure and ideological strength such that the effect of T1 exposure on unfollowing might differ for ideologically moderate vs extreme users.

## 4 DATA

The data for Studies 1 and 2 come from an **Initial Sample** of misinformation spreaders and their followers, which we describe in Section 4.1. We then created two subsets of this **Initial Sample**—a **Rate Sample** used for Study 1 (rates of unfollowing) and a **Modeling Sample** used for Study 2 (modeling predictors of unfollowing). The creation of these samples is described in sections 4.2 and 4.3. See Table 1 for how samples relate.

The **Modeling Sample** is composed of users for whom we could obtain covariate information to include in our unfollowing model and who did not leave the platform between T1 and T2. The **Rate Sample** is an active, smaller sample of followers derived from the **Modeling Sample**. We pull the users who this sample follows at T1 and T2 so we can estimate unfollowing rates.

### 4.1 Initial Sample

To construct the **Initial Sample** of spreaders and followers, we: (1) identified health misinformation rumors from PolitiFact, (2) found users who shared URLs corresponding to these rumors ('spreaders'), and then (3) collected users who followed 'spreaders' ('followers').

*Collecting Misinformation URLs and Tweets.* In March 2023 we collected all health[2] misinformation[3] rumors on PolitiFact since June 2021. We refer to the clean set of blog post URLs and tweets as Misinformation URLs and Misinformation Tweets, respectively. There were 84 such URLs and tweets.

*Identifying Eligible Spreaders.* We next identified 'Eligible Spreaders'. We conducted a search of Twitter to find all users who either (A) posted a Misinformation URL, (B) retweeted a post containing a Misinformation URL, (C) posted a Misinformation Tweet, or

---

[1]Though of course, it may be the case that not all misinformation spreaders share the same misinformation.

[2]PolitiFact categories: ['abortion', 'autism', 'coronavirus', 'drugs', 'disability', 'healthcare', 'health-check', 'public-health']

[3]PolitiFact truth values: ['pants on fire', 'false', 'mostly false']

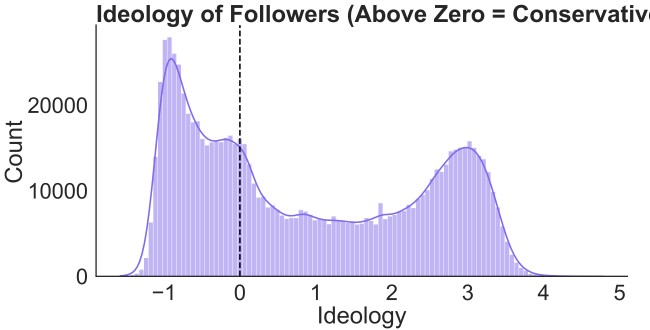

**Figure 2: Distribution of the political ideology of followers from the Modeling Sample, measured using the method from Barberá. Most followers are conservative.**

(D) retweeted a Misinformation Tweet. We denote the union of $A, B, C, D$ as Eligible Spreaders.

*Filtering Eligible Spreaders.* We applied three filters to Eligible Spreaders. First, we removed any post or retweet of a post that contained a series of debunking words. We did this to remove users who were correcting misinformation as opposed to spreading it. Debunking words (Appendix Table 2), were generated by prompting GPT-3 to expand on a list of common debunking words. Second, we selected spreaders with follower and friend counts of over ten and a follower count of less than 20K. The lower bound was applied to make sure spreaders had some activity on the platform. The upper bound on followers was partially a resource constraint (since we had to pull all of the followers), but it also allowed us to understand 'regular' users and not celebrities. After the first two filters, there were 58 Misinformation URLs and Misinformation Tweets, since not all Misinformation URLs were ever tweeted. Some of these URLs had a disproportionate number of associated tweets. Consequently, in the third filter, we used a simple greedy algorithm to retrieve 5,600 misinformation spreaders[4] while minimizing the number of spreaders that came from any specific story (Appendix Algorithm 1). These steps resulted in 5,613 misinformation spreaders (hereafter 'Spreaders'), and the rumor, or misinformation URL, with the largest number of associated spreaders made up 3.6% of the total.

*Collecting Followers.* We then pulled all followers of the spreaders at two time points: (T1) March 2023 and (T2) October 2023.

## 4.2 Modeling Sample

The participant pool for the **Modeling Sample** (used in Study 2) began from the followers and spreaders from the **Initial Sample**, and then two filters were applied. In the first filter, we restricted our analysis to only followers whose ideology could be estimated via data files provided by the first author of the 'Bayesian Ideal Point Estimation' [2] method. This is a method that estimates a user's ideology by who the user follows. Applying this restriction yielded 944,972 followers and 5,593 spreaders. As a robustness check (Appendix Figure 11), we show that relationships between *non-ideology*

---

[4]This number was based on a power analysis conducted for a relevant and concurrent project.

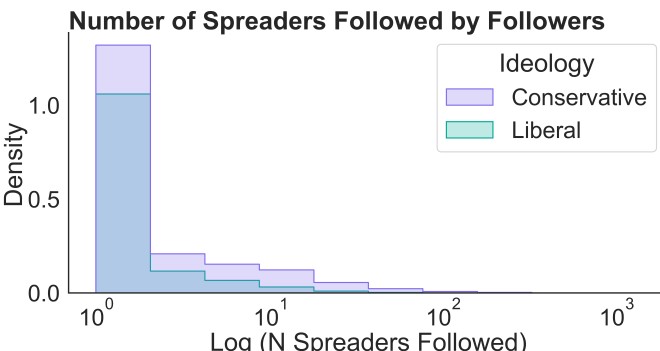

**Figure 3: Number of misinformation spreaders from the Modeling Sample followed at T1. Ideology is cut at zero using Barberá. Misinformation exposure is right-skewed.**

variables and unfollowing hold for users whose ideology we could not estimate, suggesting the structural predictors of unfollowing do not differ for users whose ideology we could not estimate. In the second filter, we removed from our analysis any follower or spreader whose basic information (e.g., followers, tweet count, friend count) could not be pulled at either T1 or T2. This can happen for multiple reasons—voluntarily exiting the platform, getting banned, etc. It is important to remove edges where either the follower or spreader left the platform since including them would distort unfollowing rates. For the **Modeling Sample**, we also had to omit users whose accounts were protected since their information could not be pulled and hence could not be included in the model. After these two filters, there were 898,701 followers, 5,334 spreaders, and 3,376,785 edges. See Appendix Table 3 for variable explanations and descriptive statistics.

Most followers in the **Modeling Sample** are conservative (Figure 2), but there is a bimodal distribution similar to the 'v-shaped' distribution of fake news exposure in 2016 [19]. Misinformation exposure is right-skewed, and conservatives follow more misinformation spreaders than liberals (Figure 3). The followers in the top 10% for the number of misinformation spreaders followed at T1

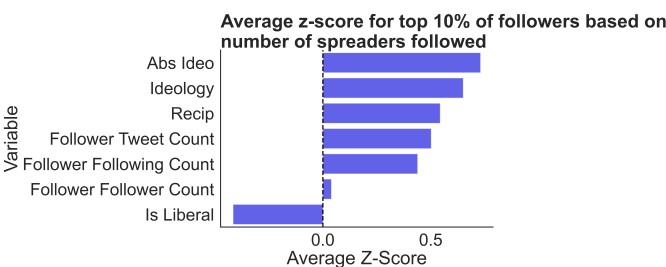

**Figure 4: Characteristics of those Modeling Sample followers who are in the top 10% for following misinformation spreaders. Ideology is a continuous measure where positive is conservative. 'Recip' is the proportion of a follower's ties to spreaders that are reciprocated.**

.

**Comparison of Twitter Unfollowing Rates**

Kirvan-Swaine et al. (2011) — 3.4%
Xu et al. (2013) — 2.5%
Maity et al. (2018) — 1.0%
Liang and Fu (2017) — 1.0%
Study 1 (2023) [Non-Misinfo Spreaders] — 0.7%
Study 1 (2023) [Misinfo Spreaders] — 0.5%

Monthly Unfollowing Rate (%)

Figure 5: Comparing unfollowing rates across studies, misinformation spreaders are unfollowed relatively infrequently.

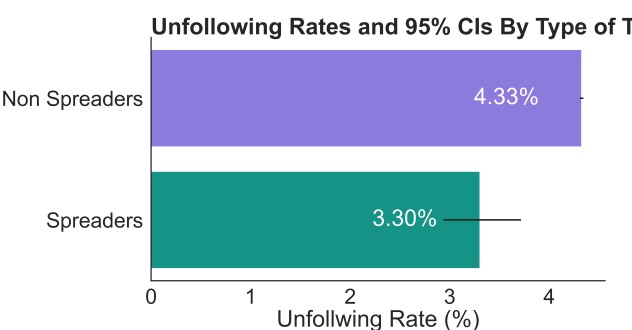

**Unfollowing Rates and 95% CIs By Type of Tie**

Non Spreaders — 4.33%
Spreaders — 3.30%

Unfollwing Rate (%)

Figure 6: In Study 1, misinformation spreaders were unfollowed less than non-misinformation spreaders.

are generally (1) ideologically extreme and (2) conservatives (Figure 4), which tracks browsing studies on selective exposure [8, 19]. There is a high reciprocity rate (69%), driven by a few spreaders. In fact, the top 10% of spreaders by reciprocal ties are responsible for 66% of the reciprocal ties; the Gini coefficient of reciprocal ties (0.79) is even larger than that of followers (0.74). This could suggest either that the most active misinformation spreaders engage in frequent 'follow-back' behavior or that the most popular misinformation spreaders are embedded in close-knit communities where reciprocation rates are high.

### 4.3 Rate Sample

To estimate unfollowing rates of misinformation spreaders and non-spreaders (Study 1), we selected a panel of 2,500 followers (**Rate Sample**) from the **Modeling Sample** and pulled who these users followed at roughly[5] the same time two points as in Study 2. The **Rate Sample** was constructed by applying several filters to followers in the **Modeling Sample**. First, we filtered followers in the **Modeling Sample** by activity level and network size (last tweet within 14 days, tweet count greater than 20, follower count between 20-20K, friends count between 20-20K) to ensure these users were active on Twitter. Second, we capped the number of friends at the 85th percentile (4,989) to further constrain network size. This was done due to resource constraints. Finally, 2,500 participants were randomly sampled from the remaining filtered pool to serve as a panel of egos.

## 5 STUDY 1: UNFOLLOWING OF SPREADERS AND NON-SPREADERS

### 5.1 Objectives

In Study 1, we analyze both (RQ1) the overall unfollowing rates of misinformation spreaders and (RQ2) if the unfollowing rate of misinformation spreaders differs from that of non-spreaders.

### 5.2 Participants

The participants started from the 2500 misinformation followers from the **Rate Sample**. See Table 1.

### 5.3 Methods

For each of these 2,500 follower 'egos', we pulled who they followed (their 'alters') at two time points—March 2023 (T1) and October 2023 (T2). We were interested in the proportion of (follower ⟹ alter) ties that were dissolved from T1 to T2 and if a tie is more likely to dissolve if the alter is a misinformation spreader. To avoid exit rates confounding unfollowing estimates—we might count a user as unfollowed if they instead deleted their account in T2—we removed all egos and alters who exited the platform because they were either suspended or their account was deleted.[6] This process yielded 2,467 egos, 2,245,645 T1 alters, and 5,087,256 T1 (follower ⟹ alter) edges. 8,052 of the initial edges were with misinformation spreaders.

### 5.4 Results

*5.4.1 RQ1: How often are misinformation spreaders unfollowed?* Misinformation spreaders are very rarely unfollowed. The unfollowing rate of misinformation spreaders was 3.3% (95% CI = [2.93%, 3.72%]), or a monthly rate of 0.52% (95% CI = [0.46%, 0.58%]). We computed confidence intervals (CIs) using the Wilson [31] method for binomial proportions. We computed monthly unfollowing rates by dividing the total unfollowing rate by $\frac{\text{study duration in days}}{30}$. This unfollowing rate is substantially lower than Twitter unfollowing rates observed in prior studies[7] (Figure 5) or what would be expected based on Kaiser et al.

*5.4.2 RQ2: Are misinformation spreaders unfollowed at a different rate than non-spreaders?* The unfollowing rate of non-misinformation spreaders (4.33%, 95% CI = [4.31%, 4.34%]) was 31% higher than the equivalent rate of misinformation spreaders (3.3%, 95% CI = [2.93%, 3.72%]). See Figure 6. Note that the 95% CIs for unfollowing rates

---

[5]Due to Twitter API instability, Study 2 started 11 days earlier and ended 4 days earlier than Study 1.

[6]We got this information by querying Twitter's Compliance endpoint and removing any accounts where the compliance status was 'suspended' or 'deleted'.
[7]Prior unfollowing studies often differ in some way (e.g: sample, exact measure, exclusion criteria) to our study, but they provide a *rough* baseline.

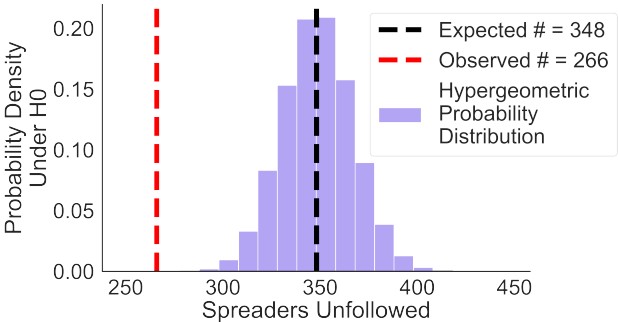

**Figure 7: Fewer misinformation spreaders in Study 1 were unfollowed than would be expected under a null hypothesis of random unfollowing ($p = 1.6 \times 10^{-6}$). We modeled the null hypothesis with a hypergeometric distribution.**

between the two groups do not overlap. (The wider confidence interval for the spreader unfollowing rate reflects the fact that there were fewer spreader edges in Study 1.)

We also tested the hypothesis that fewer spreaders were unfollowed than would be expected if unfollowing was random (or independent of whether the alter was a spreader). Under a null hypothesis of random unfollowing, we would expect the proportion of *severed* spreader ties (0.12%) to be the same as the *original* proportion of spreader ties (0.16%). But spreaders were a lower fraction of severed ties than initial ties. The null hypothesis of random unfollowing can be thought of as randomly 'sampling' users to unfollow without replacement. Then the probability of observing $k$ spreader unfollows out of $n$ total unfollows, given spreaders are $\frac{K}{N}$ proportion of initial ties, is described by the hypergeometric distribution with parameters ($N = 5,087,256, K = 8,052, n = 219,979, k = 266$):

$$P(X = k) = \frac{\binom{K}{k}\binom{N-K}{n-k}}{\binom{N}{n}}$$

and

$$P(X \le 266) = \sum_{i=0}^{266} P(X = i)$$

provides an exact p-value on the probability of observing 266 or fewer spreaders being unfollowed. That p-value is $1.6 \times 10^{-6}$ (Figure 7). We conclude that fewer spreaders were unfollowed than one would expect if unfollowing was random.

## 6 STUDY 2: PREDICTORS OF UNFOLLOWING

### 6.1 Objectives

In Study 2, we answered what predicts a user unfollowing a misinformation spreader.

### 6.2 Participants

The participants were the spreaders and followers from the **Modeling Sample**. See Table 1.

### 6.3 Methods

We modeled a follower $f$ unfollowing a misinformation spreader $s$ using cluster-robust logistic regression. The data was at an edge level (follower $f \Rightarrow$ spreader $s$). We predicted if an edge that existed at T1 (March 2023) would be dissolved at T2 (October 2023). That is, we predicted if a follower would unfollow a spreader. To account for dependencies within spreaders, we report HC1 cluster-robust errors, clustered at the spreader level.[8] See Appendix Table 4 for regression results. The covariates were the variables listed in Appendix Table 3, plus two-way interactions between our variables of interest: initial exposure (measured by number of spreaders followed at T1), ideological strength (measured by absolute value of Barberá's ideology measure), and an indicator for liberal ideology (equal to 1 if the Barberá measure is below 0). We used the *marginal-effects* R package to compute average marginal effects (AME) from our logistic regression model.

### 6.4 Results

*6.4.1 RQ3: What are the predictors of unfollowing a misinformation spreader?* See Figure 8 for non-interaction AME estimates. See Appendix Table 4 for regression results.

*Reciprocity.* Reciprocity was the biggest predictor of unfollowing. A tie being reciprocal was associated with a significantly decreased probability of unfollowing, AME = -0.0134 (95% CI = [-0.0138, -0.0131]). The unfollowing rate of non-reciprocated misinformation-spreader ties (2.14%) was 2.4 times the unfollowing rate of reciprocated misinformation-spreader ties (0.89%).

*Initial Exposure.* The results were consistent with the reversion hypothesis (**H1**) and not the inertia hypothesis (**H2**): T1 exposure (AME = 0.0028, 95% CI = [0.0026, 0.0029]) and the spreader tweeting often from T1 to T2 (AME = 0.0022, 95% CI = [0.0021, 0.0023]) were associated with unfollowing. These can both be considered measures of 'redundancy'. There was a significant interaction between T1 exposure and partisan ideology, although the effects were directionally the same (see Figure 9). T1 exposure had a larger effect for liberals (AME = 0.0040, 95% CI = [0.0035, 0.0046]) than for conservatives (AME = 0.0024, 95% CI = [0.0023, 0.0025]). Interestingly, there was also a positive interaction between T1 exposure and ideological strength (Appendix Table 4). Overall, the reversion hypothesis (high exposure at T1 is associated with high unfollowing at T2) holds on average and separately for both liberals and conservatives—though this effect is roughly 1.7 times as strong for liberals.

*Partisan Ideology.* Misinformation unfollowing shows partisan asymmetries. Liberals are more likely to unfollow than conservatives. And at higher levels of ideological strength, this gap widens. First, (**H3**) was supported: liberals were more likely to unfollow than conservatives (AME = 0.0082, 95% CI = [0.0063, 0.0101]). Second, (**H4**) was technically refuted: Averaging across the sample, there was a near-zero, *positive* AME of ideological extremity on unfollowing (AME = 0.0004, 95% CI = [0.0001, 0.0006]). However, this near-zero, average effect masked large partisan asymmetries

---

[8]We also conducted a Bayesian hierarchical logistic regression with random intercepts for spreaders, but the model yielded a convergence error. Nonetheless, the results of the Bayesian model were very similar to the results of our cluster-robust logistic regression.

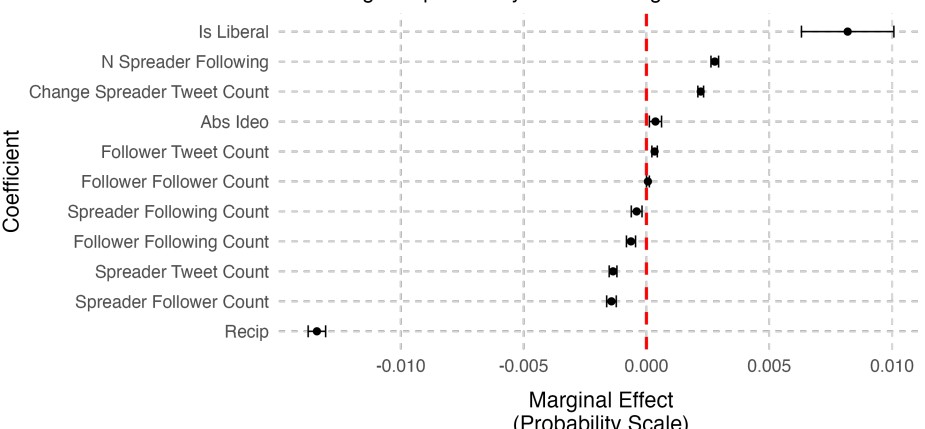

**Figure 8: Average marginal effects of predictors on the probability of unfollowing a misinformation spreader. Note that these effects are the average effects across the sample, taking into account interaction effects involving predictors.**

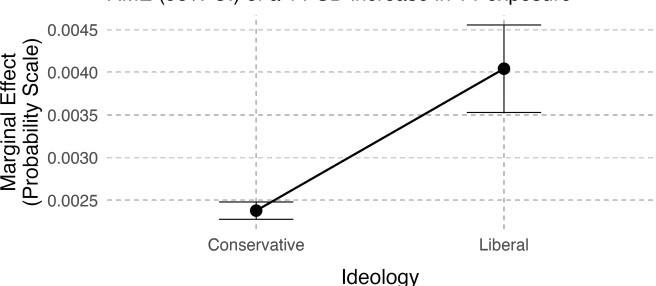

**Figure 9: The effect of T1 exposure (how many spreaders a follower followed at T1) on the probability of unfollowing differed for liberals vs conservatives. T1 exposure has a *larger* effect on unfollowing for liberal users.**

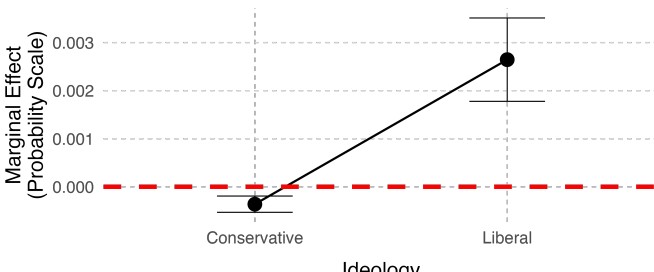

**Figure 10: The effect of ideological strength on the probability of unfollowing differed for liberals vs conservatives. Ideological strength is associated with higher unfollowing for liberals and lower unfollowing for conservatives.**

(**H5**): For liberals, an increase in ideological strength was associated with an *increase* in unfollowing (AME = 0.0026, 95% CI = [0.0018, 0.0035]). Yet for conservatives, an increase in ideological strength was associated with a *decrease* in unfollowing (AME = -0.0004, 95% CI = [-0.0005, -0.0002]). That is: Extreme liberals are *more* likely to unfollow than moderate liberals, while extreme conservatives are *less* likely to unfollow than moderate conservatives. See Figure 10.

## 7 DISCUSSION

While much is known about misinformation exposure, little is known about misinformation unfollowing. We provided a large-scale analysis of the frequency and predictors of unfollowing misinformation spreaders.

Regarding frequency, we found misinformation spreaders are rarely unfollowed. Monthly unfollowing rates are just 0.52%. Users

are also 31% more likely to unfollow *non*-misinformation spreaders than they are to unfollow misinformation spreaders. The low overall unfollowing rate suggests a role for interventions or design changes that either prevent the initial formation or encourage the dissolution of these ties. Additionally, misinformation spreaders are unfollowed less than would be expected from [12], which asked laboratory participants their hypothetical intentions to unfollow an imagined misinformation spreader. This discrepancy has two implications. First, hypothetical exercises do not capture the realities of actual unfollowing behavior, highlighting the importance of large-scale studies 'in the field'. Second, there is room for effective interventions that move individual behaviors more toward individuals' stated goals regarding their information environments.

Although rare overall, some factors did (non-trivially) predict unfollowing. We found that reciprocity had a large downward

effect on the probability of unfollowing. And likely because connections to misinformation spreaders were highly reciprocal, there was a low overall unfollowing rate. This suggests that cost-effective interventions to limit the spread of misinformation might target non-reciprocated connections, which are easier to dissolve.

Considering initial exposure, we found more evidence for a *reversion* account than an *inertia* account: Overall, higher initial exposure was associated with *higher* unfollowing and not lower unfollowing. This could be driven by redundancy (indeed, spreaders tweeting often[9] were also predictive of unfollowing) or a regression to the mean effect if high initial exposure was incidental rather than intentional. Of course, determining whether the higher rate of unfollowing was due to redundancy or regression to the mean has important implications for identifying the right messaging interventions to encourage unfollowing. Highlighting information overload costs might be more effective for the former, while messages that simply remind users of the information pollution in their networks might be more effective for the latter. Future work (e.g., interviews of individuals who unfollow misinformation producers) can help determine the mechanisms at play and inform messaging strategies.

We found partisan asymmetries. Liberals were more likely to unfollow than conservatives. Future work could explore possible mechanisms for this main effect. At more extreme ends of the spectrum, the gap between liberal and conservative unfollowing widened: Extreme liberals are *more* likely to unfollow than moderate liberals, but extreme conservatives are *less* likely to unfollow than moderate conservatives. Additionally, the reversion effect is stronger for liberals. That is, initial exposure has a larger self-correction effect for liberals than for conservatives. It is worrying that extreme conservatives are both (1) more likely to consume misinformation [8, 19] *and* (2) less likely to sever ties with those who spread it. These two dynamics suggest that misinformation exposure among conservatives is likely to stay at a high level. Our findings suggest that future work that aims to dissolve connections to misinformation spreaders should take into account the ideology of the user. On one hand, targeting left-leaning users is more likely to result in successful dissolution. On the other hand, external interventions are more needed for conservative users.

To summarize, we observed high persistence of misinformation ties and asymmetries across partisan lines. The stability of misinformation ties also points to the importance of stopping misinformation exposure in the first place.

## 8 LIMITATIONS

This work has several limitations. First, our method of identifying misinformation was 'high precision' but 'low recall'; we cannot claim our results generalize to *all* spreaders of misinformation. Second, different dynamics may hold for misinformation spreaders with more than 20K followers. We were interested in 'ordinary' and 'non-celebrity' users. Third, we refer to cases where the edge between (User A $\Rightarrow$ User B) disappeared as *unfollowing*. But API limits prevent us from determining if User B *blocked* User A or manually removed User A as a follower. We suspect these cases are less common, and they are tie dissolutions, nonetheless. Fourth, our analysis concerns one stretch of time on one platform. It is

---

[9]Though this may also be indicative of information overload.

important to replicate findings across time and platforms. However, API restrictions make this increasingly difficult. Despite these limitations, this work provides a large-scale view of misinformation unfollowing.

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

# A  APPENDIX

**Table 2: These words were generated by prompting GPT-3 with 'Here are a list of words people use when they do not agree with a statement: lie, misinformation, debunked, false. What are other words?' We used the text-davinci-003 model with a temperature of 0.7.**

| Debunking Words | | |
| --- | --- | --- |
| Baseless | Disproved | Deceit |
| Deceptive | Disputed | Distorted |
| Delusion | Erroneous | Fabricated |
| False | Falsehood | Fictitious |
| Flawed | Hoax | Implausible |
| Inaccurate | Incorrect | Lie |
| Misinformation | Misleading | Misrepresent |
| Myth | No Evidence | Not True |
| Refuted | Unfounded | Unreliable |
| Unsubstantiated | Untrue | Unverified |
| Fake | Fake News | Dubious |

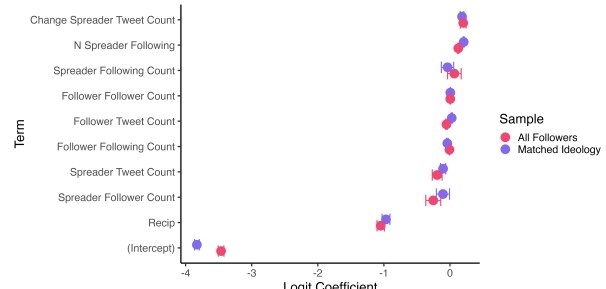

**Figure 11: We estimated a truncated version of our baseline specification (removing any ideology variables) on both the Modeling Sample and a full sample (i.e: including followers excluded from the Modeling Sample because their ideology could not be matched). Non-binary variables were z-scored. Coefficients are the effect of a +1SD change on the log odds of unfollowing. Coefficient estimates for the sample of all followers (red) were similar to those for the sample of followers whose ideology we could match (blue). Error bars are 95% CIs.**

**Table 3: Descriptive Statistics of Followers and Spreaders in the Modeling Sample. There are 898,701 followers, 5,334 spreaders, 3,376,785 edges. 'Is Reciprocal Tie' is a binary variable denoting if a tie is reciprocated. 'Is Liberal' is an indicator variable for when the ideology measure from Barberá is less than zero. The change in spreader tweet count was computed by subtracting the spreader's total count of tweets as of T2 from the equivalent tweet count in T1. In certain cases, the API returned fewer total tweets for a spreader in T2 than in T1. In these cases, we changed the 'Change Spreader Tweet Count' to zero.**

| Metric | Mean | SD | 25% | 50% | 75% |
|---|---|---|---|---|---|
| Follower Tweet Count | 18116 | 51193 | 566 | 3673 | 14984 |
| Follower Follower Count | 4311 | 208369 | 130 | 461 | 1510 |
| Follower Following Count | 2896 | 13362 | 628 | 1442 | 3107 |
| N Spreader Following | 4 | 13 | 1 | 1 | 3 |
| Spreader Tweet Count | 52311 | 97719 | 7084 | 20664 | 56644 |
| Spreader Follower Count | 1413 | 2389 | 161 | 536 | 1560 |
| Spreader Following Count | 1816 | 2223 | 387 | 962 | 2484 |
| Change Spreader Tweet Count | 4773 | 10009 | 131 | 1291 | 5018 |
| Ideology (Positive Is Conservative) | 1.65 | 1.57 | 0.06 | 2.29 | 3.0 |
| Is Reciprocal Tie | 0.69 | 0.46 | . | . | . |
| Is Liberal | 0.24 | 0.43 | . | . | . |

**Table 4: Logistic regression with HC1 cluster robust errors at the spreader level and Bayesian hierarchical logistic regression with random intercepts for spreaders. Note: Non-binary variables are z-scored so coefficients can be interpreted as the accompanying change in the log odds of unfollowing with a +1SD increase in the predictor variable.**

| | Dependent variable: | |
|---|---|---|
| | Unfollowed | |
| | Cluster-Robust Logistic Regression | Bayesian Multilevel Logistic Regression |
| | (1) | (2) |
| recip | −0.948*** (0.030) | −0.983*** (0.012) |
| is_liberal | 0.420*** (0.079) | 0.488*** (0.046) |
| follower_tweet_count | 0.027*** (0.004) | 0.030*** (0.004) |
| follower_following_count | −0.050*** (0.007) | −0.048*** (0.007) |
| follower_follower_count | 0.005*** (0.001) | 0.005** (0.002) |
| spreader_tweet_count | −0.109*** (0.021) | −0.150*** (0.017) |
| spreader_following_count | −0.032 (0.049) | 0.053* (0.028) |
| spreader_follower_count | −0.114** (0.052) | −0.206*** (0.028) |
| n_spreader_following | 0.182*** (0.006) | 0.172*** (0.005) |
| change_spreader_tweet_count | 0.177*** (0.014) | 0.223*** (0.013) |
| abs_ideo | −0.057*** (0.010) | −0.067*** (0.007) |
| is_liberal:abs_ideo | 0.264*** (0.052) | 0.313*** (0.035) |
| is_liberal:n_spreader_following | 0.174*** (0.020) | 0.166*** (0.021) |
| n_spreader_following:abs_ideo | 0.053*** (0.005) | 0.061*** (0.005) |
| Constant | −3.872*** (0.020) | −3.916*** (0.018) |
| sd(spreader) | | 0.492 |
| N | 3376785 | 3376785 |
| Note: | | *p<0.1; **p<0.05; ***p<0.01 |

**Algorithm 1** The aim is to find the minimal number of spreaders coming from a single story that allows us to retain at least a target number of spreaders.

**Require:** $T$, the target number of unique spreaders
1: Initialize $\lambda$, the maximum number of spreaders per story, to 1
2: Initialize $S$, the set of spreaders, by randomly sampling 1 user per story without replacement
3: **while** $|S| \leq T$ **do**
4:     Re-initialize $S$ as the set of spreaders derived from randomly sampling $\lambda$ spreaders per story without replacement
5:     Increment $\lambda$ by 1
6: **end while**

Received 20 February 2007; revised 12 March 2009; accepted 5 June 2009

