# OpenReview forum: "The Dynamics of (Not) Unfollowing Misinformation Spreaders"
_ACM.org/TheWebConf/2024/Conference — TheWebConf24 Oral_

### Official Review · Reviewer_YN4Q · 2023-11-11

**Novelty:** 5
**Technical Quality:** 4

**Review:**

Summary: This study examines the dynamics of unfollowing misinformation spreaders on Twitter. The authors first collect three sets of users: The initial sample, the modeling sample, and the rate sample. They then analyze the users in each sample, especially regarding the change of followers. Some interesting findings are derived, including that the misinformation spreaders are rarely unfollowed.


Strength:

1.	The authors study a crucial and novel problem - unfollowing misinformation spreaders on Twitter. Different from the conventional detection and correction of misinformation, studying these dynamics adds new value to the existing research community.

2.	For the analysis, the authors crawl large-scale modeling samples for the second research. This helps build a promising predictive model of user unfollowing behavior.


Weakness:

1.	The paper is hard to follow, and the presentation needs improvement. For instance, 1) broken logic between sentences: For “Misinformation exposure affects key decisions like compliance with health regulations and vaccine uptake intention. Due to its importance, many studies explore xxx”, the “due to its importance” seems weird. Due to the harms, instead? 2) grammar errors, “That is: ” -> “That is, ”; 3) wordy expression: “If unfollowing is common” is used multiple times in paragraph 2; 4) missing citations for “Kwak et al” and “Kaiser et al.” in Section 2.2; 5) improper presentation of bar charts: the width of bars in Figure 6 is overly large. 6) improper number presentation, “2500 -> 2,500”

2.	The motivation of the work needs to be enhanced. Particularly, what is the motivation for studying the dynamic of unfollowing? More benefits and real-world implications are required. Paragraph 2 in the introduction gives certain clues. But, a concrete example can help.


3.	For the crucial introduction section, the authors can give some findings. Otherwise, the audience knows little about the work after reading the introduction. Especially, the authors only mention the three research questions. Some critical and interesting findings are expected.


4.	I am curious about the data curation process. On page 3, the authors mention that they collect the data that dates back to June 2021. But their unfollowing rate analysis is conducted in 2023 and I think that we should conduct a short-period and timely analysis. Meanwhile, if the misinformation spreader only posts misinformation once, is it still ok to classify him or her as a misinformation spreader?


5.	The authors use GPT3 to identify the debunking words. However, the motivation and justification are not provided. Is GPT3 capable of this task? Will it provide the exhaustive list we want? More clarifications are needed.


6.	For the fourth footnote, “a relevant and concurrent project” is vague.


7.	For the Rate Sample, the authors only collect 2,500 examples, which is small. More data points for reliable analysis are required.


8.	The unfollowing rate is a key concept in the paper, but the definition is missing.

**Questions:**

Please check the weakness points.

**Reviewer Confidence:**

3: The reviewer is confident but not certain that the evaluation is correct

**Scope:**

3: The work is somewhat relevant to the Web and to the track, and is of narrow interest to a sub-community

---

### Official Review · Reviewer_y44j · 2023-11-22

**Novelty:** 6
**Technical Quality:** 6

**Review:**

Overall
Overall, I appreciate this study — it is focused, clean, and interesting.  I think my main critique is that I think the authors could have gone deeper in interpreting their results and speculating about possible reasons why they saw what they saw.  For example, maybe people don’t unfollow misinfo spreaders because they don’t realize they were spreading misinformation?  Relatedly, maybe because both T1 and T2 (the time periods the authors analyzed) could have happened after a given piece of misinformation was shared, there’s no time-salient “event” that would have sparked a follower to stop following a misinfo spreader — i.e., it’s not like they were all of a sudden made aware of the fact that one of the people they are following shared misinfo?   Maybe followers don’t unfollow spreaders because these spreaders also tend to share quality information — and perhaps they shared misinformation by accident?  Etc.  It doesn’t change the authors’ empirical results, but I think such speculative interpretations/theorizing might go a long way towards informing future work and interventions on the topic.  Right now, without alluding to these possibilities, the paper reads as if followers are very aware that they are following misinformation spreaders, that the spreaders themselves unequivocally and always have malicious intent, and that they generally produce low-quality information. If this is what the authors intend, then that’s fine, but such a depiction will invariably lead to a certain set of interventions, but perhaps not those that could have the greatest impact on improving the quality of information and discourse online.  I think explicitly probing other possible explanations could help present a broader picture, and clearly articulating them as theories/hypotheses could set the stage for future work on this important topic.

Quality
I think the study is of high quality—it is well designed and thoughtfully executed.  It focus on a narrow and scoped problem, and pursues it well.  I have a few questions/concerns about some of the methodological decisions, which are described below under Detailed comments (prepended with (--)).  These probably wouldn’t materially affect the results (namely, of study 2), but it’s hard to know for sure … they may serve as the basis of additional robustness checks if the authors haven’t already considered and/or addressed these points.

Clarity
I appreciate how the authors clearly state their hypotheses and rational/logic grounding them before diving into the data and results.  In general, I found the exposition to be clear and interesting.

Originality
This isn’t my field of expertise, but from what the authors state, the study is novel in that it explores if/how social media users update their information following/seeking behaviors after

Significance
I think this is a valuable and important contribution to the literature.  It could inspire additional research to better-understand how unfollow rates differ across those who produce more/less trustworthy information online.

Detailed comments

Introduction
* “We track how often spreaders are unfollowed from the first time point to the second time point” -> what is meant by first and second time point?

Specific hypotheses
* 3.1 — “The logic for the reversion hypothesis (H1) is that high T1 exposure would signal high redundancy” -> I think there could be many other explanations beyond redundancy.  Perhaps the content is simply not interesting to the follower anymore?
* 3.2 — ”Partisan Ideology Effect on Unfollowing” -> I don’t think this is an effect; it’s a more general assocation/relationship?  I would change the word effect, which implies a causal relationship

Data
* 4.1 — did the gpt-base debunking words filtering introduce any sort of bias into which misinfo stories were retained?
* 4.1 — ”Consequently, in the third filter, we used a simple greedy algorithm” -> I understand the details of this algo are in the appendix, but I think a bit more of an explanation here of what the purpose/goal of this algorithm was would help.  I.e., what objective is the greedy algorithm trying to optimize / what are the constraints?

Study 1
* 5.4.1 — here, we are only looking at misinformation spreaders
* (--) 5.4.2 — I’m a bit confused by the hypothesis test conducted here and some of the quantities presented.  Is the distribution showing the number of unfollows, or the number of spreaders unfollowed? I think it should be the former (given the quantity of interest is unfollows, not the number of spreaders who are unfollowed) … but in that case, given the samples aren’t iid (the same user might be responsible for multiple unfollows, so unfollows are correlated through the user), is the hypothesis test still valid here?  Is having iid samples a requirement for this test?
	* If it’s the former, I think changing the x-axis to “spreader unfollows” instead of “spreaders unfollowed” might help

Study 2
* (--) 6.3 — why did the authors use cluster robust standard errors instead of / not in addition to fixed or random effects to model the follower and/or spreader?
	* It seems like modeling the follower is important in order to account for user-level idiosyncrasies that might be associated with following/unfollowing behavior?
* In general, I think it’s worth reframing the results from “effects” to “associations” — see earlier feedback on this.  “Effect” makes it sound like there’s a causal relationship, but I don’t think that can be inferred from the study design, which is observational in nature

**Questions:**

Please see above for questions.

**Reviewer Confidence:**

3: The reviewer is confident but not certain that the evaluation is correct

**Scope:**

4: The work is relevant to the Web and to the track, and is of broad interest to the community

---

### Official Review · Reviewer_vtTh · 2023-11-23

**Novelty:** 6
**Technical Quality:** 3

**Review:**

This study tackles an important problem of understanding the dynamics of unfollowing misinformation spreaders. After collecting tweets on health misinformation and the followers of the spreaders, they asked three research questions on the unfollowing frequency and predictors of unfollowing behaviors. They found that misinformation spreaders are less likely to be unfollowed. Also, the factors related to the unfollowing behaviors were identified as redundancy and ideology.

The research questions are clear and well-motivated. This paper is generally written well, and I agree that answering these questions could contribute to the research on responsible web. While there are several in-text references missing numbered citations, this study provides a fair amount of related works and discussions of the findings based on the previous studies.

However, this paper has a key weakness regarding the validity of the findings.
-I think unfollowing rates could be confounded by popularity measures such as the number of followers, which is not properly controlled in the analysis. This study applied several filtering criteria to make the minimum and maximum of the spreaders and general users the same, yet the distribution could be different. Authors could present the distribution of confounding attributes that could affect unfollowing rates. An analysis method that takes those factors into account could be employed.
-The method of inferring political orientation should be elaborated. Also, since the method is old, its accuracy should be reported on a dataset with a similar distribution of the analysis target.

**Questions:**

NA

**Reviewer Confidence:**

3: The reviewer is confident but not certain that the evaluation is correct

**Scope:**

4: The work is relevant to the Web and to the track, and is of broad interest to the community

---

### Official Review · Reviewer_81Fo · 2023-11-23

**Novelty:** 6
**Technical Quality:** 4

**Review:**

The paper investigates to which extent users on Twitter unfollow accounts that have spread misinformation. For this, the authors first aggregated a list of misinformation spreaders by filtering Twitter accounts that have shared a post with a link to a health-related misinformation URL. These misinformation URLs are sourced from PolitiFact. The paper aggregates ~5k misinformation spreaders, followed by ~900k users (for which the authors could identify the political affiliation).

To check the unfollowing rate, the paper samples 2500 users and looks at who these accounts follow at two points: March 2023 and October 2023.
The paper shows that the unfollowing rate per month is low at 0.52%. The paper also investigates what features are predictive of unfollowing.

The paper clearly states the hypotheses that are tested, clearly describes the methodology, and performs statistical tests.

Overall, the results are interesting, and I was genuinely curious to learn what the unfollowing rate is. In terms of how generalizable are the results, the analysis has several limitations, some of which are acknowledged by the authors:
- The analysis is done over a biased set of users. The seed list of misinformation URLs is sourced from PolitiFact.

 - The paper looks at one-shot misinformation spreaders — users that shared one misinformation URL. When evaluating news organizations, most journalistic agencies (MediaBiasFactCheck) are looking at **repeated** spreaders of misinformation. So, I am wondering how the results would look if the paper instead only used seed users that repeatedly spread misinformation (e.g., more than ten posts).

- Organized information operations are making use of bots to spread misinformation. The paper looks at the unfollowing behavior of a sample of 2500 users, but we do not know to which extent these are legitimate people or bots.

After rebuttal:
- I had hoped that authors would provide some numbers of unfollowing rate if we look at users who repeatedly spread misinformation. This is not a very complicated analysis. The authors just said that they are not looking at this. I think understanding the unfollowing rate of repeated spreaders of misinformation is very important and would provide a more comprehensive and reliable picture of unfollowing. If many of the users in the dataset are one-shot spreaders and they generally post truthful information, then the unfollowing rate presented is misleading (it could be much higher if we look at repeated spreaders).

- the excuse for not detecting bots is also not strong; some services might not currently work, but there is enough literature in the space to just check some features to test for highly suspicious accounts, even if we do not detect all of them.

**Questions:**

What is the unfollowing rate if we only look at users who repeatedly spread misinformation  (e.g., more than 10 posts) ?

**Reviewer Confidence:**

4: The reviewer is certain that the evaluation is correct and very familiar with the relevant literature

**Scope:**

4: The work is relevant to the Web and to the track, and is of broad interest to the community

---

### Official Review · Reviewer_kBYZ · 2023-11-25

**Novelty:** 5
**Technical Quality:** 6

**Review:**

This work tries to evaluate the behaviour of users when following misinformation spreaders.

The work is interesting and rigorous in analysing behaviour on twitter and using metrics, hypotheses and statistical tests to evaluate it. This is a very good paper in my opinion and I did not find reasons for which it would fall short of being accepted.

I have though questions related to the impact of it: the wordings seem to show that you are evaluating settings that are strictly North American in your work. If I am correct and even if not, how do you think your work could be applied to different countries (even outside the westernalised settings)? How do you think the English language is affecting the generalisability of the approach? Aside that these could be interesting elements of your discussion, it would be interesting future work to expand the work in these directions.

The only actual issue I see is the ethical aspects of the work: the use of data that can identify individuals (though public) as well as the implications of your results have ethical aspects that have not been considered.

**Questions:**

Please see above my questions on implications, impact, and ethics

**Ethics Review Description:**

The work does not mention any ethical aspect and, although tweets are public, the authors have used information that can identify individuals.

**Ethics Review Flag:**

Yes

**Reviewer Confidence:**

2: The reviewer is willing to defend the evaluation, but it is likely that the reviewer did not understand parts of the paper

**Scope:**

4: The work is relevant to the Web and to the track, and is of broad interest to the community

---

### Decision · Program_Chairs · 2024-01-22

**Decision:**

Accept (Oral)

**Comment:**

We appreciate the authors' work in engaging with reviewers at the rebuttal stage. We support the area chair's recommendation (below) for accepting the submission. We ask the authors to address concerns raised by the reviewers in the camera-ready version. We particularly underline the ethics flag raised by reviewer kBYZ and we expect the authors to update the manuscript for camera-ready.

"The authors seek to determine whether users unfollow misinformation spreaders on social media. The reviewers expressed appreciation for the study's approach, findings, and presentation. They also had some concerns about the analysis (such as repeated vs one-time spreaders, confounders). The authors seemed willing to address concerns in a revision, and I therefore recommend acceptance."